# Update on the Classification and Pathophysiological Mechanisms of Pediatric Cardiorenal Syndromes

**DOI:** 10.3390/children8070528

**Published:** 2021-06-22

**Authors:** Giorgia Ceravolo, Tommaso La Macchia, Caterina Cuppari, Valeria Dipasquale, Antonella Gambadauro, Celeste Casto, Maria Domenica Ceravolo, Maricia Cutrupi, Maria Pia Calabrò, Paola Borgia, Gianluca Piccolo, Alessio Mancuso, Remo Albiero, Roberto Chimenz

**Affiliations:** 1Unit of Emergency Pediatric, Department of Human Pathology in Adult and Developmental Age “Gaetano Barresi”, University of Messina, “G. Martino” Policlinic, 98124 Messina, Italy; giorgiaceravolo@gmail.com (G.C.); caterina.cuppari@polime.it (C.C.); dipasquale.va@gmail.com (V.D.); gambadauroa92@gmail.com (A.G.); celestecasto@libero.it (C.C.); maria.domenica.ceravolo@gmail.com (M.D.C.); cutrupimaricia@gmail.com (M.C.); Alessiomancuso10@gmail.com (A.M.); 2Unit of Cardiology, Department of Clinical and Experimental Medicine, University of Messina, “G. Martino” Policlinic, 98124 Messina, Italy; tommasolamacchia90@virgilio.it; 3Unit of Pediatric Cardiology, Department of Human Pathology in Adult and Developmental Age “Gaetano Barresi”, University of Messina, “G. Martino” Policlinic, 98124 Messina, Italy; mpcalabro@unime.it; 4Department of Neurosciences, Rehabilitation, Ophthalmology, Genetics, Maternal and Child Health, University of Genoa, 16132 Genoa, Italy; paolaborgia92@gmail.com (P.B.); dottorpiccolo.gianluca@gmail.com (G.P.); 5Pediatric Neurology and Muscular Diseases Unit, IRCCS Istituto Giannina Gaslini, 16147 Genoa, Italy; 6Department of Cardiology, Sondrio General Hospital, 23100 Sondrio, Italy; albiero@panvascular.com; 7Unit of Pediatric Nephrology, and Rheumatology with Dialysis, Department of Human Pathology in Adult and Developmental Age “Gaetano Barresi”, University of Messina, “G. Martino” Policlinic, 98124 Messina, Italy

**Keywords:** cardiorenal syndrome, pathophysiology, chronic kidney disease, pediatric age, heart failure, renal function, child, kidney injury, treatment, management

## Abstract

Cardiorenal syndrome (CRS) is defined as a disorder resulting from the abnormal interaction between the heart and kidney, in which acute or chronic dysfunction of one organ may lead to acute and/or chronic dysfunction of the other. The functional interplay between the heart and kidney is characterized by a complex bidirectional symbiotic interaction, regulated by a wide array of both genetic and environmental mechanisms. There are at least five known subtypes of CRS, based on the severity of clinical features and the degree of heart/renal failure. The fourth subtype (cardiorenal syndrome type 4 (CRS4)) is characterized by a primary chronic kidney disease (CKD), which in turn leads to a decreased cardiac function. Impairment of renal function is among the most important pathophysiological factors contributing to heart failure (HF) in the pediatric age group, and cardiovascular complications could be one of the most important causes of mortality in pediatric patients with advanced CKD. In this context, a loss of glomerular filtration rate directly correlates with both the progression of cardiovascular complications in CRS and the risk of HF. This review describes the interaction pathways between the heart and kidney and the recently identified pathophysiological mechanisms underlying pediatric CRS, with a special focus on CRS4, which encompasses both primary CKD and cardiovascular disease (CVD).

## 1. Introduction

The heart and kidneys interact through several intricate cellular and subcellular processes and molecular pathways [1]. These interactions represent the pathophysiological basis for a clinical condition defined as cardiorenal syndrome (CRS), which is characterized by acute or chronic dysfunction of either the kidney or the heart, leading to potential acute and/or chronic dysfunction of the other organ [2]. CRS may lead to dual-organ dysfunction and the worst outcome for the patient. A clear definition of these interactions and a better knowledge of organ cross-talk are required to establish an effective treatment for individuals affected with CRS. This is particularly important in pediatric patients in whom cardiovascular disease (CVD) can be regarded as a major risk of premature death in the context of advanced chronic kidney disease (CKD) [3].

In recent years, next-generation sequencing technologies, including exome and mRNA sequencing studies, have shed new light on the complex molecular mechanisms underlying CRS [4,5]. This has led to a better understanding of the intricate subcellular mechanisms underlying these conditions and promoting the risk of both CRS and CVS, including several regulatory networks and transcriptional factors, such as the interferon regulatory factor 1 [6]. 

In recent years, studies based on next-generation sequencing and omics-related sciences have revealed an expanding complexity underlying rare pediatric disorders associated with chronic organ failure. The molecular dissection of these conditions and the underlying genetic and/or immune-mediated possible causes may explain the frequent multisystemic, metabolic, and neurological abnormalities associated with these rare pediatric disorders [7,8,9,10]. Many novel molecular factors have been identified with consequent benefits in terms of refining clinical phenotypes, valuable prognostic information, detailed imaging studies, and targeted therapies for the affected children [11,12,13,14,15].

## 2. Definition of CRS

In 2004, the Working Group of the National Heart, Lung, and Blood Institute proposed the first definition of CRS. CRS was described as a consequence of abnormal functional connections between the kidneys and other circulatory compartments, leading to increased circulating blood volume and exacerbating the symptoms of heart failure (HF) and CVD progression. This initial definition was extremely cardiocentric, thus not explaining the real complexity underlying the interplay between the two organ systems. In 2008, the Acute Dialysis Quality Group proposed a new classification in a Consensus Conference. This classification divided CRS into two key sets, cardiorenal and renocardiac CRS, according to the primum movens of disease (cardiac or renal). Both cardiorenal and renocardiac CRS were then subsequently organized into five subtypes based on disease acuity and according to the beginning and duration of the organ dysfunction (Table 1). The new classification overcame some of the initial ambiguity in defining CRS and helped clinicians to aim for a personalized approach, in both diagnosis and treatment, for each patient. However, in clinical practice, it can be challenging to identify the initial pathophysiological events of CRS [16].

## 3. Epidemiology of CRS

The lack of univocal terminology in the research literature and the difficulties to reach a broadly accepted definition for this condition represent an unsolved issue, which complicates epidemiological studies of CRS. The use of uniform terminology is needed to: (i) better compare data across disease subtypes and different patients; (ii) understand the epidemiology underlying CRS and investigate demographic and age- and gender-related factors and their contribution to the disease initiation and progression; (iii) dissect the underlying pathomechanisms of kidney dysfunction and HF and to fully characterize the specific subtype of heart/kidney disorder according to the individual clinical scenario [17]. Based on the data from the principal agency of the U.S. Federal Statistical System (NCHS), the death rates in the general population were 0.31 per 1000 population for American children aged 1–19 years in 2008. Pediatric patients with kidney disease (ex: end-stage renal disease (ESRD) on dialysis) have a 10–30-fold higher risk of all-cause cardiovascular mortality [18]. According to US Renal Data System data, the mortality rates in children aged 0–19 years with ESRD were 35.6 (dialysis) and 3.5 (transplant) per 1000 patient-years at risk from 2006 to 2008. In addition, children on dialysis live 40–50 years less, while transplanted patients live nearly 20–25 years less than the US general population with the same age and race.

## 4. Pathophysiology of CRS

The pathophysiological interactions and feedback mechanisms between the cardiovascular and renal systems are complex and bidirectional. There are three major mechanisms beyond the development and the evolution of cardiorenal and renocardiac dysfunction: hemodynamic alterations, neurohormonal mechanisms, and other factors such as biochemical perturbations and structural changes [19]. These interactions between the heart and kidney extend across several pathways and can result in the dysfunction of one organ or both. The most known hemodynamic pathway includes fluid overload, prerenal hypoperfusion, and the activation of the renin–angiotensin–aldosterone axis (RAAS). The impairment of the RAAS system leads to fluid and salt reabsorption, which in turn leads to worsening heart failure. This has been implicated in several rare pediatric disorders associated with increased venous congestion and a rise in either blood or cerebrospinal fluid pressure [20,21,22,23]. Renal hemodynamic changes are the basis for the augmented level of serum creatinine and venous congestion, which might be the cause of accelerated renal impairment in this clinical setting. Several neurohormonal pathways are operative in CRS, such as the common compensatory mechanisms of heart failure, activation of the RAAS axis, activation of the sympathetic nervous system (SNS), imbalance of reactive oxygen species [24], and chronic inflammation. Other pathophysiological factors include multiple mechanisms that contribute to the development and/or worsening of cardiovascular diseases, such as the development of systemic and local acute and chronic inflammation, imbalance immune responses, bone–mineral and acid–base abnormalities, anemia, and cachexia. The clinical significance of each mechanism varies from patient to patient and the setting of circumstance.

### 4.1. Hemodynamic Mechanisms in CRS

The conventional hypothesis for CRS focuses on the hemodynamic adaptations of the kidney to heart failure as an initial insult. CRS results from the heart’s inability to generate forward flow leading to hypoperfusion of the kidneys (low flow) [25]. Renal hypoperfusion is initially associated with relative preservation of the glomerular filtration rate (GFR) through renal autoregulatory mechanisms, including an increase in efferent arteriolar resistance and glomerular capillary pressure, vasoconstriction, and vasodilation of afferent and efferent arterioles. Renal venous hypertension and congestion, enhanced renal fibrogenesis, and, finally, loss of renal function result from these factors. The kidneys receive 25% of cardiac output (because they are a low-resistance circuit), and even with a large reduction in renal blood flow, they are able to preserve the GFR. However, in the case of severe reduction of the cardiac output, the renal compensatory mechanisms are insufficient, with a subsequent decline in GFR. In particular, the hypoperfusion of the kidneys triggers the baroreceptors with activation of the RAAS axis, which leads to renal vasoconstriction, increased proximal tubular sodium, and water reabsorption, eventually worsening heart failure and limiting organ perfusion. These mechanisms lead to an elevation in serum creatinine levels and may finally result in renal tubular hypoxia and acute tubular necrosis. Thus, renal dysfunction in heart failure causes increased cardiac congestion and right-atrial and central venous pressure, which may contribute to fluid overload and increased venous return, creating a vicious cycle. These processes contribute to the acute condition, and the presence of comorbidities (such as diabetes mellitus) may worsen the clinical status [26].

### 4.2. Neurohormonal Mechanisms in CRS

Neurohormonal mechanisms play a significant role in CRS pathophysiology. The regulation of the heart and kidneys involves a complex interaction between neurohormones, the sympathetic nervous system (SNS), and filling status. The neurohormonal pathway is linked to RAAS and SNS activation, which, together with hemodynamic mechanisms, lead to systemic vasoconstriction, salt and water retention, and preservation of GFR. The concomitant activity of SNS can exacerbate RAAS-mediated fluid overload and increase systemic vasoconstriction, which, in patients with cardiac dysfunction, can lead to a congestive state with peripheral edema. RAAS axis activation, in these patients, leads also, indirectly, to angiotensin II activation and the formation of oxygen reactive species (ROS) [27,28]. Compensatory mechanisms finally lead to increased renal and cardiac fibrogenesis, as well as heart and kidney dysfunction.

### 4.3. Genetic Mechanisms in CRS

Rare pediatric disorders associated with chronic organ failure may be involved in several molecular pathways, including the regulation of transcription, translation, and post-translational processes at the subcellular level. These disorders include a variety of monogenic and polygenic/genetically complex and/or environmental conditions with expanding clinical differential diagnosis, molecular heterogeneity, and associated (underlying) disease mechanisms [29,30,31,32].

Several other molecular pathways have been implicated in CRS, including chronic inflammation, comorbidities [33,34] such as systemic atherosclerosis, mineral disorders, cardiorenal anemia, malnutrition, diabetes mellitus, and arterial hypertension, all of which contribute to the onset and/or progression of cardiac and/or renal dysfunction [35,36]. Cardiac and renal dysfunction can be further accelerated by the development of systemic and local inflammation with altered immune responses, as well as by the presence of cardiorenal anemia and mineral disorder. In addition, mineral disorders, such as hyperphosphatemia, can further exacerbate chronic or acute renal damage and/or cardiac illness by remodeling and/or fibrogenesis, as well as the subsequent loss of organ function [19].

## 5. CRS Classification

CRS includes all the disorders involved in the bidirectional interactions between the heart and kidneys, in which the dysfunction of one organ can significantly impair the function of the other. CRS is divided into two main categories based on the initial insult and into five subcategories according to the direction of the effect and the acuity of the initiating insult (acute or chronic) [4,5] Table 1. CRS type 1 reflects an acute worsening of cardiac function affecting the kidneys, type 2 encompasses chronic cardiac disorders worsening the kidney’s function, type 3 comprises acute renal abnormalities causing acute heart failure, type 4 reflects a chronic kidney disorder leading to decreased cardiac function and heart failure, and, finally, type 5 describes a systemic insult leading to cardiorenal dysfunction.

### 5.1. CRS Type 1

CRS type 1 (acute cardiorenal) is characterized by the fast deterioration of cardiac function, which leads to acute renal injury (AKI). Type 1 CRS is relatively prevalent. It is usually present in the setting of acute decompensated heart failure (ADHF), often after an ischemic (acute coronary syndrome, cardiac surgery complications) or nonischemic heart disease (valvular disease, pulmonary embolism), which may be divided into four subtypes: hypertensive pulmonary edema with preserved left ventricular systolic function, acutely decompensated chronic HF, cardiogenic shock, and predominant right ventricular failure. 

*Pathophysiology*. *Hemodynamic mechanisms*. In the presence of ADHF, hemodynamic processes play a prominent role in CRS type 1, resulting in decreased renal artery flow and, as a result, a decrease in GFR. Different hemodynamic profiles have been proposed: in “cold” pattern patients, the predominant hemodynamic change is a drop in effective circulating fluid volume, whereas in “wet” pattern patients, there is a significant increase in central venous pressure. *Nonhemodynamic mechanisms*. CRS type 1 has been linked to SNS and RAAS activation, chronic inflammation, and an imbalance in the percentage of reactive oxygen species (ROS)/nitric oxide (NO) generation. The beginning of AKI has an impact on the prognosis of HF in CRS type 1, but early detection of AKI is difficult [37]. The preservation of both cardiac and renal function should be prioritized.

### 5.2. CRS Type 2

Chronic anomalies in heart function (e.g., chronic congestive HF) contribute to progressive kidney failure in CRS type 2 (chronic cardiorenal syndrome).

*Pathophysiology. Hemodynamic mechanism.* The most important pathophysiological processes of CRS type 2 are renal hypoperfusion and persistent elevations in renal venous pressure. *Nonhemodynamic mechanisms*: Maladaptive activation of the RAAS axis and the SNS, as well as a chronic inflammatory state, are important pathophysiological causes of renal disease progression. Structure injury, such as glomerulosclerosis and tubulointerstitial fibrosis, is precipitated by intrarenal oxidative stress and proinflammatory signaling. Chronic HF and renal illness can coexist for a long time, making it difficult to determine which of the two disease states is predominant (or secondary). Even a small drop in glomerular filtration rate (GFR) results in a markedly poor prognosis, increasing the mortality rate in these individuals, and it can be used as a marker of cardiovascular disease severity [38]. Diabetes mellitus, advanced age, hypertension, and acute and coronary syndrome are also other useful indicators of a poor prognosis [39]. 

### 5.3. CRS Type 3

When AKI leads to the development of acute HF, CRS type 3 (acute renocardiac syndrome) ensues. It is less prevalent than type 1 CRS and has not been thoroughly researched in the literature. AKI can impact the heart directly or indirectly, and through numerous distinct pathways, it can cause an abrupt cardiac event [40].

*Pathophysiology. Hemodynamic mechanisms*. Renal dysfunction can cause severe pathophysiological derangement, which can lead to heart damage. Oliguria can cause sodium and water retention, resulting in fluid overload and the development of hypertension, pulmonary edema, and myocardial injury. *Nonhemodynamic mechanisms*. Electrolyte abnormalities that cause arrhythmias and cardiac arrest, metabolic acidosis that causes vasoconstriction, coronary artery disease, and left ventricular dysfunction can be also associated with CRS type 3. Renal ischemia can cause myocardial inflammation and apoptosis on its own [41]. Furthermore, baroreceptor and intrarenal chemoreceptor activation is triggered by changes in systemic and renal hemodynamics. These conditions have a negative impact on cardiac outcomes.

### 5.4. CRS Type 5

CRS type 5 (secondary CRS) is characterized by the presence of cardiac and renal dysfunction in the same patient, and it can occur in a variety of clinical settings as a result of acute or chronic systemic illnesses (e.g., sepsis, hepatorenal syndrome, and Fabry’s disease). Type 5 CRS has a scarcity of information in the literature. It is obvious that some systemic diseases affect both organs at the same time, and that the malfunction of one might have an impact on the other. The most common cause of CRS type 5 is sepsis, which causes cell ultrastructural changes and organ failure. Sepsis can cause AKI and severe cardiac depression at the same time. Many mediators and pathways have been implicated in the pathogenesis of sepsis-induced cardiac depression; however, the exact etiopathogenetic mechanism remains unknown.

*Pathophysiology*. *Hemodynamic mechanisms*. Pathophysiological abnormalities in sepsis-related CRS type 5 are influenced by the sepsis’s systemic effects and direct cross-talk between the injured heart and kidney. Despite good systemic hemodynamics, microcirculation is frequently involved in the early stages of sepsis. With dilatation and decreased ejection fraction, both the left and right ventricles can be affected in septic cardiomyopathy. There are obvious changes in intraparenchymal blood flow in sepsis-associated AKI, independent of systemic hemodynamic abnormalities due to the septic process. *Nonhemodynamic mechanisms*. The pathophysiology of septic cardiomyopathy does not appear to include myocardial blood flow or oxygen consumption. The role of proinflammatory mediators and complement factors in the development of cardiac involvement during sepsis has been suggested. Sepsis can influence the autonomic nervous system (ANS), the RAAS, and the hypothalamus–pituitary–adrenal axis (HPA) independently, affecting cardiac and/or renal function in numerous, separate processes. Several physiological and molecular alterations occur in both tissues with combined heart and kidney dysfunction, as in sepsis. Activation and elevation of cytokines (TNF-a and IL-6), as well as leukocytes (macrophages, neutrophils, and lymphocytes) in the heart and kidney during sepsis, has been well documented.

## 6. CRS Type 4 

### 6.1. Definition of Type 4 CRS

CRS type 4 (chronic renocardiac syndrome) is defined by cardiovascular involvement (e.g., impaired cardiac function, ventricular hypertrophy, diastolic dysfunction, higher risk of adverse cardiovascular events) in patients with primary CKD (e.g., chronic glomerular disease) [42]. Figure 1 shows the strong relationship between CKD and cardiovascular disease. GFR is a powerful and independent predictor of cardiovascular morbidity and mortality, with severe cardiac events accounting for over half of all deaths in individuals with CKD. Other predictors of severity are: chronic inflammation markers, insulin resistance, hyperhomocysteinemia, and lipid dysmetabolism.

### 6.2. Epidemiology of Type 4 CRS 

CKD is a major public health problem. It accounts for at least 11% of the U.S. adult population [43]. The National Center for Health Statistics reported in 2008 a mortality rate of 0.31 per 1000 in the general US pediatric population (from 1 to 19 years old) [3]. In contrast, the US Renal Data System (USRDS) reported a mortality rate of 35.6 (dialysis) and 3.5 (transplant) per 1000 patient-years at risk for children with ESRD (aged 0–19) from 2006 to 2008 [44]. Children with ESRD have a severely reduced life expectancy, especially dialysis patients who live 40–50 years shorter than an age- and race-matched US population [44], while transplant patients live around 20–25 years less. The American Heart Association classified pediatric CKD patients as having the highest risk of developing CVD [45]. Children with CKD have the highest cardiovascular risk of any pediatric population, and in these patients, cardiovascular death occurs mostly for cardiac arrest followed by arrhythmia, cardiomyopathy, and cerebrovascular disease, while myocardial infarction is rarely reported. The prevalence of cardiac arrest in the youngest age group (0–4 years) is 5–10 times higher than in other pediatric age groups [46]. In children with advanced CKD, atherosclerosis is commonly noted, due to a unique mix of conventional and uremia-related risk factors. This evidence explains why young patients who get CKD as children are more likely to develop symptomatic CAD.

### 6.3. Risk Factors of Type 4 CRS

Children with CKD have a high prevalence of conventional cardiovascular risk factors (e.g., hypertension, dyslipidemia, diabetes) [47]. Modern information on cardiovascular risk factors in this population is provided by the Chronic Kidney Disease in Children (CKiD) study, which evaluated 586 children (1–16 years of age) with stage 2 to 4 CKD. On enrolment, hypertension was present in 54% of participants. In spite of the assumption of an antihypertensive therapy, in 48%, BP was not adequately controlled. Hyperphosphatemia and hypertension themselves contribute to vascular calcification and consequent pressure overload. In the CKiD study, dyslipidemia was found in 45% and obesity in 15%, while insulin resistance and hyperinsulinemia were present in 19% and 9%, respectively [48]. It is worth mentioning that almost half of the patients in the CKiD study had an association with conventional risk factors. The risk of traditional factors increases with the progression of CKD, and children on dialysis and transplanted have the highest risk for CVD. Traditional risk factors [49], unfortunately, are insufficient to completely explain the high prevalence of cardiac death in children from 0 to 19 years old on dialysis or transplanted [50,51,52]. In these patients, the complications leading to death may be linked also to uremia-related risk factors [6]. A cross-sectional investigation of all pediatric maintenance hemodialysis patients (aged 0.7–18 years, *n* = 656) revealed that 38% have anemia, 63% have serum phosphorus >5.5 mg/dL, and 55% have a calcium–phosphorus product >55 mg^2^/dL^2^ [53]. Despite the above findings, we need to further uncover the link between traditional and/or uremia-related risk factors and CVD mortality.

### 6.4. Pathophysiology of Type 4 CRS

*Pathophysiology. Hemodynamic mechanisms.* Ischemic heart disease is accelerated by CKD, which also contributes to pressure and volume overload, resulting in left ventricular hypertrophy (LVH). Congestive HF is worsened by volume overload central to CKD, with underlying anemia of chronic illness and the occurrence of hemodialysis arteriovenous fistulas being typical contributing factors. Nonhemodynamic mechanisms. Traditional risk factors of CVD, such as hemodynamic derangements, hypertension, and RAAS and SNS activation, do not completely explain the high rates of HF in CKD patients [54]. As a consequence, CKD-related risk factors are receiving more attention.

According to recent experimental and clinical studies, CKD leads to an accumulation of uremic toxins [55]. 

Huang et al. recently demonstrated in an experimental study that a uremic molecule, known as high phosphate (HP), contributes to CKD-associated HF [6]. HP is crucial in the impairment of myocardial energy metabolism function by inducing alterations in the transcriptional factor interferon regulatory factor 1 (IRF1)–peroxisome proliferator-activated receptor-gamma coactivator 1 alpha (PGC1a) signaling pathway.

Indeed, HP upregulates IRF1 through acetylation of the H3K9 histone, which mediates HP-induced PGC1α downregulation by binding directly to its promoter region. PGC1α is downregulated in the hearts of CKD mice, implying that it mediates myocardial energy metabolism dysfunction and leads to CKD-related HP. On the other hand, genetic knockdown of IRF1or PGC1α restoration significantly improves HP-induced changes. 

Huang et al. demonstrated that myocardial energy metabolism dysfunction induced by the IRF1-PGC1α axisplays a crucial role in the pathogenesis of type 4 CRS and suggested that hyperphosphatemia management or targeted intervention on HP-mediated IRF1 elevation and PGC1α downregulation could represent a future treatment able to reduce cardiovascular risk in CKD patients [6].

The activation of the local renin–angiotensin system has recently been discovered to have a vital role in the course of illnesses. Both the heart and kidney’s local renin–angiotensin systems are linked to cardiovascular disease and kidney disease, respectively. Angiotensin II production in the tissues is just as significant as circulating angiotensin II. These local RAAS have both autocrine (cell to cell) and paracrine (cell to cell) actions. As a result, the RAAS is more complex than a simple blood volume and blood pressure control mechanism. Target-organ damage is mediated by AT1 activation at the local level, which includes remodeling, endothelial dysfunction, and collagen deposition, resulting in fibrosis. Furthermore, angiotensin II is synthesized locally, and the amount of angiotensin II produced locally increases in direct proportion to the severity of HF. In HF patients, angiotensin II synthesis increases sodium reabsorption, causes systemic and renal vasoconstriction, cardiac hypertrophy, apoptosis, and changes in the interstitium.

### 6.5. Markers of Cardiovascular Involvement of Type 4 CRS

Evidence of early cardiovascular abnormalities can be found even in patients with mild kidney dysfunction. Common findings are abnormalities of the left ventricular (LV) (e.g., LV hypertrophy (LVH) and LV dysfunction) regions, large artery structural alterations (e.g., stiffness, increased intima-medial thickness of the carotids), and coronary and cardiac valve calcifications [56]. These findings are strong, independent markers of cardiac involvement and atherosclerosis and can be used as predictors of cardiac morbidity and mortality, not only in the general populations but also in children and young adults with CKD, and are usually worse in children on maintenance dialysis [57].

### 6.6. Treatment of Type 4 CRS

The correct treatment of CRS type 4 necessitates a comprehensive and combined approach, considering the pathological mechanisms. Currently, the treatment of type 4 CRS is mainly based on the correction of the traditional and nontraditional cardiovascular risk factors and on the prevention of CKD progression. Once diagnosed, heart disease should be treated according to current HF treatment guidelines, such as those published by the European Society of Cardiology.

#### 6.6.1. Pharmacological Therapy

In patients with mild to moderate renal impairment, angiotensin-converting enzyme inhibitors (ACEIs) have been shown to be effective in improving the cardiovascular survival rate of patients with myocardial disease [58,59].

Moreover, ACEIs help dialysis patients avoid arrhythmic events thanks to their action on neurohormonal stimulation, ventricular remodeling, and hemodynamics. In addition, Pun et al. [60] demonstrated that ACEIs or angiotensin receptor blockers (ARBs) and beta-blockers (BBs) have a beneficial effect in ESRD patients, leading to increased survival rate after cardiac arrest. In hypertensive patients, ARBs can help to minimize inflammation and oxidative stress, suppress the RAAS, and reduce cardiovascular events [61].

Little is known in the literature on the use of ARBs in dialysis patients to reduce cardiac mortality. Candesartan was tested in a small randomized study in dialysis patients. Candesartan was shown to reduce cardiovascular events and fatal arrhythmias in dialysis patients in a small, randomized trial [62], though the small number of events limited the importance of the latter result. Moreover, Suzuki et al. [63] demonstrated that ARBs would help minimize cardiovascular injuries in dialysis patients. However, the value of ARB usage for the prevention of cardiovascular accidents and sudden death in dialysis patients requires larger trials to be performed. 

It has been demonstrated that after myocardial infarction or heart failure, BBs reduce the risk of cardiovascular death. BBs minimize cardiac risk in CAD patients with or without CKD, as shown in the Bezafibrate Infarction Prevention study [64]. A total of 114 dialysis patients enrolled in a randomized study by Cice et al. [65] received carvedilol or placebo with a substantial decrease in cardiovascular mortality and a trend toward a decrease in sudden death in the carvedilol group. According to other trials, hemodialysis patients who were given BBs had lower overall and cardiovascular mortality [66].

In conclusion, there is no definitive clinical guideline for the best mode of coronary intervention in CRS type 4 patients at this time.

#### 6.6.2. Dialytic Therapies

The theory of “cardioprotective dialysis” states that advances in dialysis can enhance hemodynamic stability, minimize oxidative and inflammatory stress, and result in the more effective removal of low and middle toxins. New technologies include the use of new biomaterials that can help minimize inflammatory reactions and increase membrane performance, as well as machines with well-integrated safety, therapy effectiveness, performance, and monitoring functions.

##### Dialysis Technological Improvement Goals

Patient-monitoring improvements during hemodialysis (HD) can increase safety and tolerability, which is critical. The ultrafiltration rate and conductivity have been regulated to modify blood volume and plasma refilling utilizing biofeedback devices. In contrast to conventional HD, Peritoneal dialysis (PD) may be able to escape the hemodynamic instability that comes with normal and rapid ultrafiltration.

However, some evidence indicates that although PD has a lower mortality rate than HD in the first 1–2 years, it has a higher mortality rate after that. On the other hand, other registry data contradict this conclusion [67,68]. 

##### Dialysis Modalities

A Cochrane meta-analysis to assess the effects of different hemodialysis membrane materials in patients with ESRD showed that biocompatible membranes were associated with a reduction in the beta-2-microglobulin (β2M) level, an increase in albumin concentration, and improvement in Kt/V, but mortality was not affected [69].

House et al. [70] found no difference in lipids and homocysteine levels between high-flux polysulfone membranes and low-flux polysulfone membranes in a controlled trial. In an observational study, Chauveau et al. [71] discovered that high-flux membranes were associated with improved 2-year survival.

Several studies have linked “hemodiafiltration” or “hemofiltration” care to improve blood pressure regulation, improve β2M and phosphate clearance, decrease intradialytic hypotension or arrhythmia, reduce inflammation and oxidative stress, and a lower hospitalization rate [72,73].

##### Dialysis Late Complications

Chronic kidney disease–mineral and bone disorder (CKD–MBD) is a dialysis consequence that is especially noteworthy in children, as it can interfere with growth and cardiovascular health. Furthermore, all-cause mortality rates are greater for children on dialysis: at least 30 times higher than the overall pediatric population, and much higher than the transplant population.

## 7. Conclusions 

CRS patients have a higher risk of death and morbidity than those with each disease entity alone. The heart and kidneys interact bidirectionally and interdependently through several mechanisms. A better understanding of the interactions between these organs is important for his practical clinical implications. Patients with CKD have the highest cardiovascular risk of any pediatric demographic, and dialysis is a primary factor linked to poor outcomes in children with ESRD. These data suggest that long-term dialysis should be avoided in these patients, with preemptive transplantation as the best management strategy when feasible. When compared to long-term dialysis, successful transplantation can considerably alleviate uremia-related risk factors and, most importantly, enhance life expectancy by 20–30 years. More research is needed to better understand the pathogenetic process and best therapies in this pediatric population.

## Figures and Tables

**Figure 1 children-08-00528-f001:**
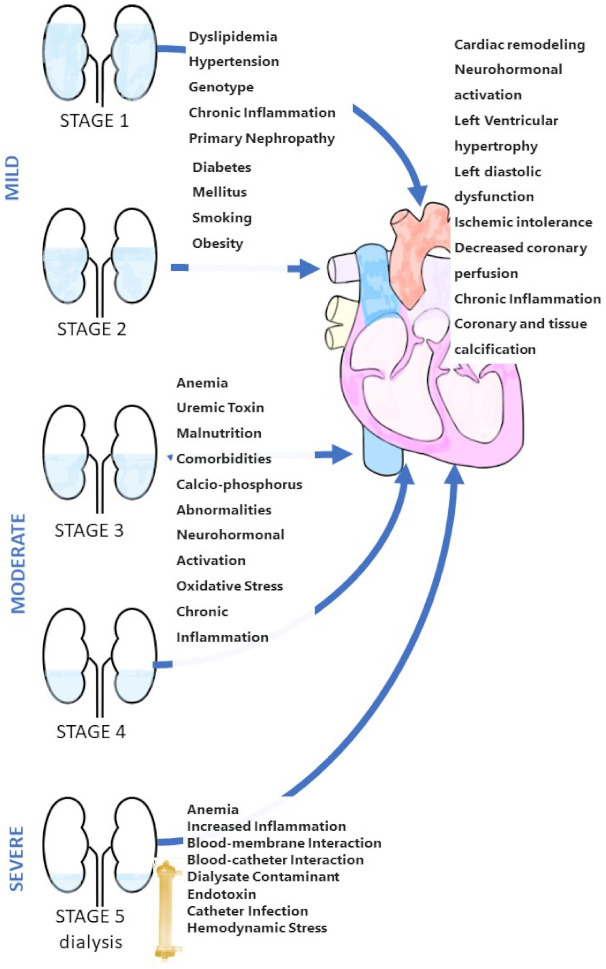
Schematic of CRS type 4.

**Table 1 children-08-00528-t001:** CRS Classification.

CRS Type	Denomination	Description
Type 1	Acute cardiorenal	Heart failure leading to acute kidney disease
Type 2	Chronic cardiorenal	Chronic heart failure leading to acute kidney disease
Type 3	Acute renocardiac	Acute kidney disease leading to acute heart failure
Type 4	Chronic renocardiac	Chronic kidney disease leading to heart failure
Type 5	Secondary	Systemic disease leading to heart and kidney failure

## Data Availability

The data presented in this study are available on request from the corresponding author.

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
