# Peer review of "Update on the Classification and Pathophysiological Mechanisms of Pediatric Cardiorenal Syndromes"

_children, 2021, doi:10.3390/children8070528_

Round 1
Reviewer 1 Report
The authors address the pathophysiological mechanism of pediatric cardiorenal syndrome (CRS) and argue the interaction between heart and kidney for clinical implications. Also, They focus on the pathogenesis of CRS4 characterized by cardiovascular involvement in primary CKD patients and explain the pathophysiology and treatment for the disease. The manuscript is well written and very important for the readers in the field. The reviewer has just one concern as follows.
Recently, it is known that the local renin-angiotensin system activation plays critical role in the progression of diseases. The local renin-angiotensin system in both heart and kidney are associated with cardiovascular disease and CKD, respectively. The authors should discuss the local renin-angiotensin system involved with CRS4.
Author Response
The authors address the pathophysiological mechanism of pediatric cardiorenal syndrome (CRS) and argue the interaction between heart and kidney for clinical implications. Also, They focus on the pathogenesis of CRS4 characterized by cardiovascular involvement in primary CKD patients and explain the pathophysiology and treatment for the disease. The manuscript is well written and very important for the readers in the field. The reviewer has just one concern as follows:
Point 1: Recently, it is known that the local renin-angiotensin system activation plays critical role in the progression of diseases. The local renin-angiotensin system in both heart and kidney are associated with cardiovascular disease and CKD, respectively. The authors should discuss the local renin-angiotensin system involved with CRS4.
Response 1: Under the subtitle “6.4. Pathophysiology of type 4 CRS” we added the following new text:
“Recently, it is known that the local renin-angiotensin system activation plays critical role in the progression of diseases. The local renin-angiotensin system in both heart and kidney are associated with cardiovascular disease and CKD, respectively. The generation of angiotensin II at the tissue level is as important as circulating angiotensin II. These local RAAS exert autocrine (cell-to-same cell) and paracrine (cell-to-different cell) effects. As a result, the RAAS is more complex than a simple pathway controlling blood volume and blood pressure.
The local activation of AT1 mediates target-organ damage including remodeling, endothelial dysfunction, and collagen deposition resulting in fibrosis. Moreover, angiotensin II is also synthesized locally, and local angiotensin II production is increased proportionally to the severity of HF. In HF patients, local and systemic angiotensin II production leads to increased sodium reabsorption, systemic and renal vasoconstriction, myocardial hypertrophy, apoptosis and alterations in the interstitium.”
thank you for your point, Please see the attachment.

Reviewer 2 Report
the goal to characterize cardio-renal syndrome is good but gets murky with the incorrect English, plurals, improper acronyms. The authors should try to organize around main thoughts: hemodynamic, hormonal, genetic, fluid perturbations of each type of CRS. The section on dialytic therapies could be divided into dialysis modalities and complications and technological improvement goals.
Author Response
Point 1: the goal to characterize cardio-renal syndrome is good but gets murky with the
incorrect English, plurals, improper acronyms. The authors should try to organize around
main thoughts: hemodynamic, hormonal, genetic, fluid perturbations of each type of CRS.
The section on dialytic therapies could be divided into dialysis modalities and complications
and technological improvement goals.
Response 1: As suggested we corrected all the English words, including plurals, other grammar errors an acronyms. In addition, for each type of CRS, we organized the text around main thoughts on pathophysiology: “hemodynamic and non-hemodynamic mechanisms”.
For CRS type 1 this text was added:
Pathophysiology. Hemodynamic mechanisms play a major role in
CRS type 1 in presence of ADHF leading to decreased renal arterial
flow and a consequent fall in GFR. Different hemodynamic profiles have
been proposed: in “cold” pattern patients, reduction in effective
circulation fluid volume represents the main hemodynamic change,
while there is a marked increase in central venous pressure in “wet”
pattern patients. Non-hemodynamic mechanisms proposed as involved
in type 1 CRS include SNS and RAAS activation, chronic inflammation
and imbalance in the proportion of reactive oxygen species (ROS)/nitric
oxide (NO) production.
For CRS type 2 this text was added:
Pathophysiology. Hemodynamic mechanisms. Renal hypoperfusion
and chronic increases in renal venous pressure represent the most
important pathophysiological mechanisms of type 2 CRS. Non-
hemodynamic mechanisms: Important pathophysiological triggers of
renal disease progression includemaladaptive activation of the RAAS
axis and the SNS, as well as a chronic inflammatory state. Intrarenal
oxidative stress and proinflammatory signaling precipitate structural
injury, including glomerulosclerosis and tubulointerstitial fibrosis.
For CRS type 3 this text was added:
Pathophysiology. Hemodynamic mechanisms. As renal function
declines, it can result in significant pathphysiological derangement,
leading to cardiac injury. Oliguria can lead to sodium and water retention
with consequent fluid overload and development of volume overload,
hypertension, pulmonary edema and myocardial injury. Non-
hemodynamic mechanisms: CRS type 3 can be also associated with:
electrolyte disorders contributing to arrhythmias and cardiac arrest;
metabolic acidosis producing vasoconstriction; coronary artery disease;
left ventricular dysfunction. Renal ischemia itself may also precipitate
cardiac inflammation and apoptosis [41].. Moreover, in response to
systemic and renal hemodynamic changements, baroceptor and
intrarenal chemoceptors lead to SNS and RAAS activation. These
conditions have a deleterious effect on cardiac outcomes.
For CRS type 5 this text was added:
Pathophysiology. Hemodynamic mechanisms. Pathophysiological
changes in sepsis related CRS 5 depend on systemic effects of the
sepsis itself, and also, from direct cross-talk between the damaged
heart and kidney. In early stages of sepsis microcirculation is often
initially involved despite normal systemic hemodynamics. In septic
cardiomyopathy, both the left and right ventricle can be injured with
dilation and decreased ejection fraction. In sepsis associated AKI, there
are clear changes in intra-parenchymal blood flow independent of
systemic hemodynamic changes linked to the septic process. Non-
hemodynamic mechanisms. Myocardial blood flow and oxygen
consumption do not seem involved in pathophysiology of septic
cardiomyopathy. Pro-inflammatory mediators and complement factors
have been proposed as crucial actors in the development of cardiac
involvement during sepsis. Sepsis is able to affect the autonomic
nervous system (ANS), RAAS and hypothalamus-pituitary gland-adrenal
gland axis (HPA) independently which can impact, in several and
distinctive steps, cardiac and/or renal function. It’s clear that during
combined heart and kidney dysfunction, as in sepsis, several cellular
and molecular changes occur in both tissues. Activation and induction of
cytokines (TNF-a and IL-6) and leukocytes (macrophages, neutrophils
and lymphocytes) is well documented both in heart and kidney during
sepsis.
For CRS type 4 this text was added:
Pathophysiology. Hemodynamic mechanisms. CKD independently
accelerates ischemic heart disease and contributes to pressure and
volume overload, leading to left ventricular hypertrophy (LVH).
Congestive HF is exacerbated by volume overload central to CKD with
underlying anemia of chronic disease and the presence of hemodialysis
arteriovenous fistulae being common contributing factors. Non-
hemodynamic mechanisms.
As suggested, the section on dialytic therapies was divided into dialysis modalities and complications and technological improvement goals.
thank you for your points, Please see the attachment.

Round 2
Reviewer 2 Report
GFR is mistyped several times. Heart(h) is misspelled. Worse is written as worst. Pathophysiology is misspelled. Circulation should be circulating. Space between include and maladaptive.
Author Response
Points: GFR is mistyped several times. Heart(h) is misspelled. Worse is written as worst. Pathophysiology is misspelled. Circulation should be circulating. Space between include and maladaptive.
Response 1: As suggested we corrected all the mistyped and misspelled words, including GFR, heart, worse, pathophysiology, circulating and spaces.
Please see the attachment.
